# Working strokes produced by curling protofilaments at disassembling microtubule tips can be biochemically tuned and vary with species

**Lucas E Murray[1], Haein Kim[1], Luke M Rice[2,3], Charles L Asbury[1,4]***

[1]Department of Physiology and Biophysics, University of Washington, Seattle, United States; [2]Department of Biophysics, UT Southwestern Medical Center, Dallas, United States; [3]Department of Biochemistry, UT Southwestern Medical Center, Dallas, United States; [4]Department of Biochemistry, University of Washington, Seattle, United States

**Abstract** The disassembly of microtubules can generate force and drive intracellular motility. During mitosis, for example, chromosomes remain persistently attached via kinetochores to the tips of disassembling microtubules, which pull the sister chromatids apart. According to the conformational wave hypothesis, such force generation requires that protofilaments curl outward from the disassembling tips to exert pulling force directly on kinetochores. Rigorously testing this idea will require modifying the mechanical and energetic properties of curling protofilaments, but no way to do so has yet been described. Here, by direct measurement of working strokes generated in vitro by curling protofilaments, we show that their mechanical energy output can be increased by adding magnesium, and that yeast microtubules generate larger and more energetic working strokes than bovine microtubules. Both the magnesium and species-dependent increases in work output can be explained by lengthening the protofilament curls, without any change in their bending stiffness or intrinsic curvature. These observations demonstrate how work output from curling protofilaments can be tuned and suggest evolutionary conservation of the amount of curvature strain energy stored in the microtubule lattice.

*For correspondence:
casbury@uw.edu

Competing interest: The authors declare that no competing interests exist.

## Editor's evaluation

This important and technically sophisticated work advances our understanding of force production by depolymerizing microtubules with implications for the generation of forces that segregate chromosomes during cell division. The authors present compelling evidence for their mechanistic conclusions. This work will be of interest for cell biologists and biophysicists interested in cell division and force production by biopolymers.

## Introduction

Microtubules are filamentous polymers central to the active transport of cargoes in cells. While they often serve as passive tracks along which dynein and kinesin motors move, these filaments can also drive motility directly. Dynamic microtubules in the mitotic spindle transport chromosomes during cell division by shortening while their disassembling tips remain coupled via kinetochores to the chromosomes (*Desai and Mitchison, 1997*; *Inoué and Salmon, 1995*; *McIntosh et al., 2010*). Dynamic microtubules also generate force to properly position the mitotic spindle and the nucleus within cells

**eLife digest** Dividing cells duplicate their genetic information to create identical pairs of chromosomes, which then need to be equally distributed to the two future daughter cells. In preparation, each chromosome in a pair is pulled towards its final location by hollow tubes of proteins known as microtubules. To create this tugging force, the microtubule acts like a winch: the extremity attached to the chromosome gradually shortens by losing its building blocks. However, it is not clear how the microtubule can keep its grip on the chromosome while also 'falling apart' in this way.

A possible explanation could stem from the way that microtubules are built, and from how they fall apart. Each tube is composed of rows of building blocks, called 'protofilaments'. As the microtubule shortens, the protofilaments first curl outwards before crumbling apart; this creates a curling action that could 'hook' the chromosome and pull on it as the microtubule shortens. This theory remains difficult to test however, in part because scientists lack ways to alter the properties of curling protofilaments in order to dissect how they work.

Murray et al. aimed to fill that gap by using a technique they have previously developed, and which allows them to capture how much force curling protofilaments can apply on their environment. This approach uses an instrument known as laser tweezers to measure the pressure that microtubules exert on attached beads. With this assay, Murray et al. were able to investigate whether microtubule 'strength' is linked to protofilament length, a property that varies between species and in response to magnesium. The experiments revealed that adding magnesium not only lengthens protofilament curls but also increases the work generated from curling. In addition, they showed that yeast protofilaments create longer curls with more force compared to bovine microtubules. Together, these findings demonstrate that it is possible to fine-tune the force exerted by protofilaments on their environment by controlling their length. This knowledge could be helpful to scientists investigating the role of microtubules in cell division.

Certain cancer drugs already target microtubules in order to stop rogue cells from multiplying. However, serious side-effects often emerge because these compounds also interfere with microtubule-based processes essential for healthy cells. By better understanding how protofilaments 'pull' on chromosomes, it may become possible to design targeted approaches to stop cell division but preserve the other fundamental roles that microtubules play in the body.

(*Carminati and Stearns, 1997*; *Dogterom et al., 2005*; *Kozlowski et al., 2007*; *McIntosh et al., 2010*; *Nguyen-Ngoc et al., 2007*). These microtubule-driven movements are powered by GTP hydrolysis. GTP is incorporated into the assembling polymer tip and then hydrolyzed, depositing energy into the GDP-tubulin lattice. The stored lattice energy is released during disassembly and can be harnessed to generate pulling force. Thus microtubules, like dynein and kinesin motors, convert chemical energy into mechanical work (*McIntosh et al., 2010*). How they do so remains poorly understood.

Two distinct classes of mechanism could explain how disassembling microtubule tips generate pulling force: the biased diffusion and conformational wave mechanisms (*Asbury et al., 2011*). According to biased diffusion-based models, a tip-coupler such as the kinetochore undergoes a thermally driven random walk along the microtubule surface that is biased at the tip, due to the affinity of the coupler for the microtubule. If the affinity of the coupler for the microtubule is sufficiently high and if its diffusion is sufficiently fast, then the coupler can remain persistently associated with the disassembling tip, where it will experience a thermodynamic force in the direction of disassembly (*Hill, 1985*). The effect is analogous to capillary action that pulls fluids into narrow channels. Biased diffusion of a key kinetochore element, the Ndc80 complex, has been observed directly on microtubules in vitro (*Powers et al., 2009*).

By contrast, force generation in conformational wave-based models depends on structural changes at disassembling microtubule tips. During disassembly, individual rows of tubulin dimers called protofilaments curl outward from the tip before breaking apart, creating a wave of conformational change that propagates down the long axis of the microtubule (*Kirschner et al., 1974*; *Mandelkow and Mandelkow, 1985*). These curling protofilaments are proposed to physically hook the kinetochore and pull against it to drive motility (*Koshland et al., 1988*). Prior work showed that the amount of mechanical strain energy released by curling protofilaments is more than sufficient to account for

kinetochore motility (*Driver et al., 2017*). However, whether kinetochores specifically harness any of this strain energy remains unclear, owing in part to the lack of methods for modifying mechanical or energetic properties of protofilament curls.

Many prior studies have established that added magnesium profoundly affects the dynamics of microtubules in vitro, altering the rates of switching between tip growth and shortening (*O'Brien et al., 1990*), accelerating tip disassembly (*Martin et al., 1987*), and lengthening the protofilament curls at disassembling tips (*Mandelkow et al., 1991*). Binding of magnesium to acidic residues in the disordered C-terminal tail of tubulin is implicated in magnesium-dependent acceleration of disassembly (*Fees and Moore, 2018*; *Sackett et al., 1985*; *Serrano et al., 1984a*; *Weisenberg, 1972*). Faster disassembly by itself might explain why magnesium also lengthens protofilament curls, because it implies a faster rate of curling (i.e. that GDP-tubulins are losing their lateral bonds and curling outward more quickly; *Tran et al., 1997*). However, magnesium might also stabilize the longitudinal bonds within protofilament curls, thereby reducing the rate at which the curls break. To disentangle magnesium's effects on curling and breakage rates, a systematic examination of curl contour length as a function of disassembly speed is required.

Previously, we developed an assay for measuring forces and displacements generated by curling protofilaments (*Driver et al., 2017*) based on earlier pioneering work (*Grishchuk et al., 2005*). In our 'wave' assay, the curling protofilaments push laterally against a microbead tethered to the microtubule wall, thereby generating a brief pulse of bead motion against the force of a feedback-controlled laser trap. We show here that the sizes of these pulses – and the mechanical work energy that can be harnessed from them – are substantially increased by the addition of millimolar levels of magnesium. By measuring wave pulses after proteolytic cleavage of the β-tubulin C-terminal tail, we show that magnesium enlarges the pulses independently of its acceleration of disassembly, indicating that magnesium directly stabilizes the longitudinal bonds within protofilament curls. We also demonstrate that pulses generated by yeast tubulin are larger than those generated by bovine brain tubulin. A simple mechanical model shows that both the magnesium- and species-dependent changes in pulse energy can be explained solely by increasing the contour lengths of protofilament curls, without changing their intrinsic flexural rigidity or curvature. The conservation of protofilament flexural rigidity and stored lattice strain suggest that these biophysical properties are crucial to microtubule function in cells.

## Results

### Measuring outward curling of protofilaments from bovine brain microtubules

We previously measured the mechanical and energetic properties of protofilaments as they curled outward from recombinant yeast-tubulin microtubules (*Driver et al., 2017*). In our wave assay, a laser trap applies force against the curling protofilaments, via beads tethered to the microtubule lattice through a single His$_6$ tag on the C-terminus of β-tubulin (*Johnson et al., 2011*). Linkage through a single β-tubulin C-terminal tail creates a strong, flexible tether approximately 36 nm in length, which probably helps to avoid interference between the tethered bead and the curling protofilaments (*Driver et al., 2017*). To extend our approach to untagged mammalian brain tubulin, we modified the assay by introducing anti-His beads pre-decorated sparsely with the recombinant His$_6$-tagged yeast tubulin into chambers containing coverslip-anchored microtubules growing from free bovine brain tubulin. The decoration density of yeast tubulin on the beads was kept very low, around one tubulin per bead, by limiting the amount of anti-His antibody on the beads (see Materials and methods). The bead-linked yeast tubulin was incorporated into the assembling bovine microtubules, resulting in beads tethered to the sides of the filaments (*Murray et al., 2022*; *Figure 1a*). As in our previous work (*Driver et al., 2017*), the low density of antibody on the beads ensured that most beads were tethered by a single antibody. Continuous tension, directed toward the plus end, was applied to a microtubule-tethered bead using feedback control. The tension pressed the bead against the microtubule lattice at a secondary contact point and suppressed Brownian motion, which facilitated tracking the bead with high spatiotemporal resolution. The microtubule plus end was then severed with laser scissors to induce disassembly (*Franck et al., 2010*). As the disassembling tip passed the secondary contact point, protofilament curls pushed laterally on the bead, causing it to rotate about

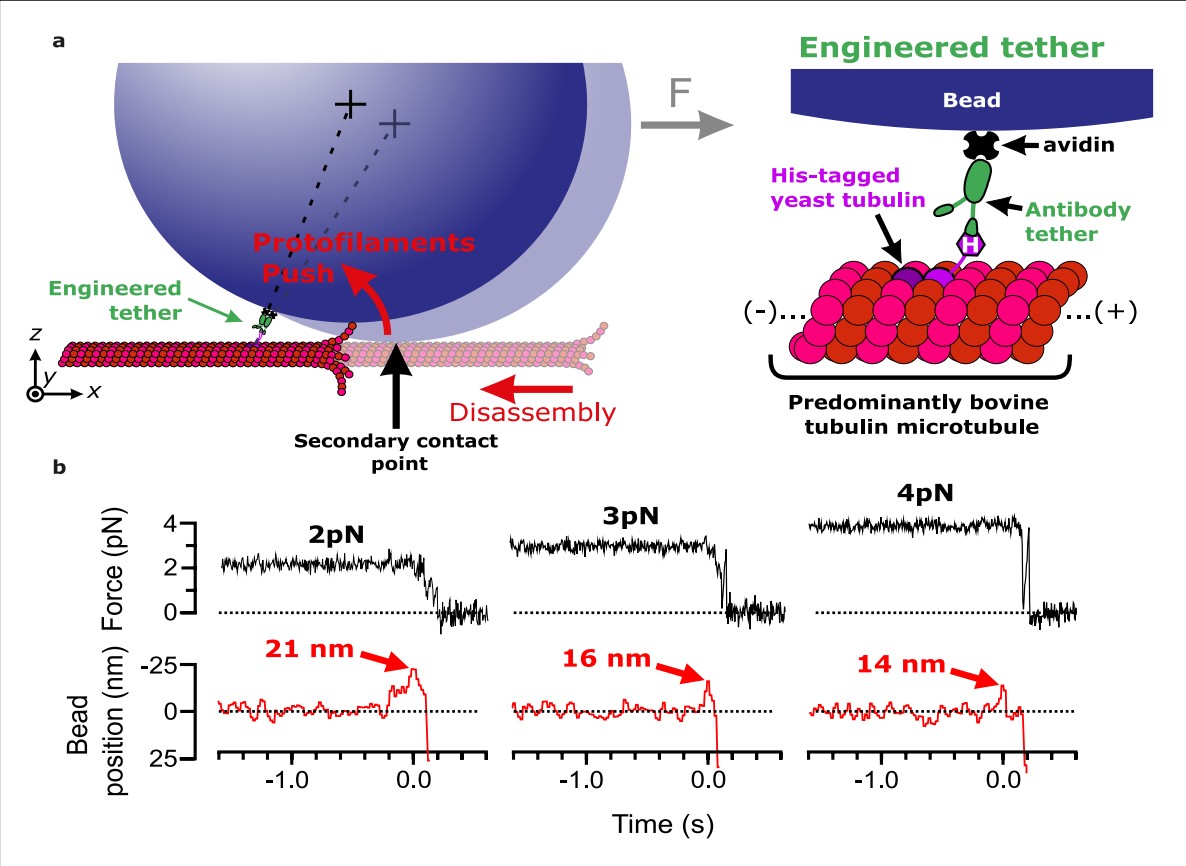

**Figure 1.** Measuring pulses of movement generated by protofilaments curling outward from the tips of disassembling bovine microtubules. (**a**) Schematic of the wave assay: a bead is tethered to the microtubule lattice via an engineered tether composed of recombinant His$_6$-tagged yeast tubulin, a biotinylated anti-penta-His antibody, and streptavidin. Tethering by a single anti-penta-His antibody is ensured by keeping the density of antibodies on the beads very low. Using a laser trap, the bead is tensioned toward the (+)-end, pressing it against the microtubule lattice at a secondary contact point. The stabilizing GTP cap is trimmed off the microtubule with laser scissors to initiate disassembly. Curling protofilaments at the disassembling microtubule tip form a conformational wave that pushes laterally on the bead, causing it to rock back about its tether. This rocking action produces a pulse of bead movement against the force of the laser trap. (**b**) Records of force (black) and bead position (red) versus time for three different bead-microtubule pairs. As the trapping force on the bead was increased, pulse heights decreased, consistent with spring-like behavior of the protofilament curls.

The online version of this article includes the following figure supplement(s) for figure 1:

**Figure supplement 1.** Features of the pulses of bead motion generated by curling protofilaments.

**Figure supplement 2.** The bead and tether form a leverage system that amplifies protofilament curling motion.

its tether. This rotation produced a brief (100–400 ms) pulse of bead movement against the force of the laser trap, which was followed by bead detachment after further disassembly released the tether (*Figure 1b*). The pulses were parameterized by their amplitude relative to the baseline bead position (*Figure 1—figure supplement 1*), which is directly related to the lateral height that the protofilament curls project from the surface of the microtubule lattice (*Figure 1—figure supplement 2*) (*Driver et al., 2017*).

At 2 pN of trapping force, 59% of disassembly events yielded measurable pulses, with a mean amplitude of 19.2±2.7 nm. At higher forces, pulse amplitudes became smaller (*Figure 1b*), consistent with spring-like elasticity of the curling protofilaments, as we previously observed for yeast tubulin protofilament curls (*Driver et al., 2017*). Pulse amplitudes generated by bovine microtubules were smaller than those we measured previously from yeast microtubules at identical force levels (e.g. 19.2±2.7 vs 51±7 nm on average at 2 pN) (*Driver et al., 2017*). This observation suggests that bovine protofilament curls might be shorter than yeast curls, consistent with reports that disassembly products released from mammalian brain microtubules are shorter than their yeast-derived counterparts

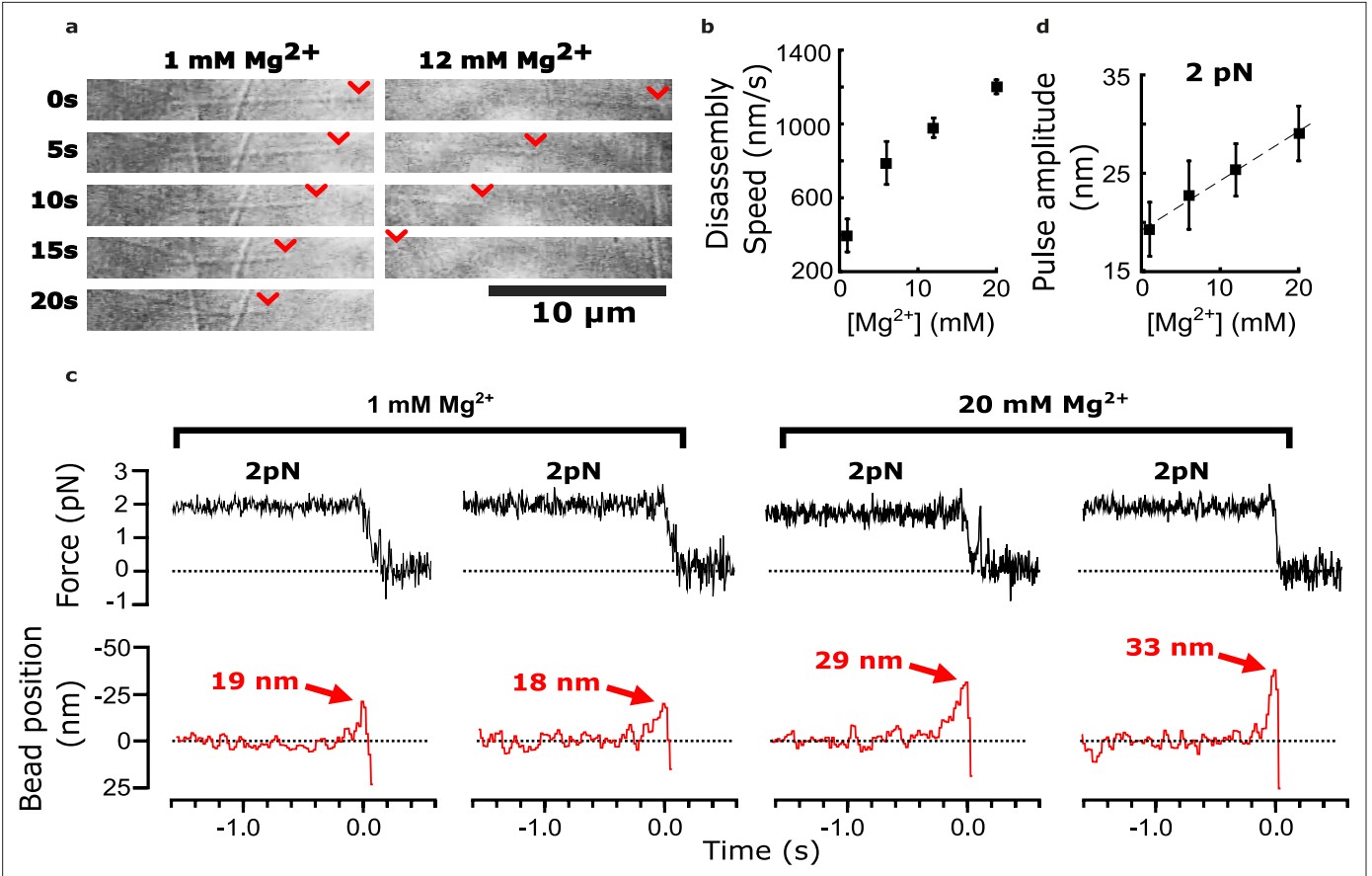

**Figure 2.** Added magnesium increases disassembly speed and pulse amplitude. (**a**) Time-lapse differential interference contrast images of individual microtubules disassembling in the presence of 1 or 12 mM magnesium. Arrowheads (red) indicate locations of disassembling tips. (**b**) Mean disassembly speed plotted against magnesium concentration. Error bars represent 95% confidence intervals, defined as ± (*t*·SEM), where *t* is drawn from Student's *t*-distribution (with $v=N-1$ degrees of freedom and N=34–51 samples per mean). (**c**) Records of force (black) and bead position (red) versus time for four bead-microtubule pairs, at two different magnesium concentrations. Pulse amplitudes were larger at the higher magnesium level. (**d**) Mean pulse amplitudes across four different magnesium concentrations, 1, 6, 12, and 20 mM. Error bars represent 95% confidence intervals (defined as in (**b**), with $v=N-1$ degrees of freedom and N=25–40 samples per mean). Data in (**c**) and (**d**) were collected at 2 pN trap force.

The online version of this article includes the following source data for figure 2:

**Source data 1.** Individual pulse amplitudes and disassembly speeds measured using bovine microtubules across different magnesium levels.

(*Howes et al., 2018*). Nevertheless, our findings confirm that pulses from bovine microtubules can be reliably measured using our modified wave assay.

## Adding magnesium enlarges the pulses generated by curling protofilaments

Divalent cations have long been known to affect tubulin self-association (*Nogales et al., 1995*; *Olmsted and Borisy, 1975*; *Weisenberg, 1972*) and influence microtubule dynamics (*Rosenfeld et al., 1976*; *Weisenberg, 1972*). These effects occur partly through interactions of magnesium ions with the unstructured C-terminal tails of tubulin (*Fees and Moore, 2018*; *Serrano et al., 1984b*) and with the exchangeable and non-exchangeable nucleotide binding sites (*Lee and Timasheff, 1975*). Early cryo-electron microscopy of disassembling microtubules showed that magnesium lengthens protofilament curls at disassembling tips (*Mandelkow et al., 1991*). Based on these prior observations, we predicted that pulses recorded in our wave assay might become larger and more energetic with added magnesium.

As previously observed (*Fees and Moore, 2018*), we found that adding magnesium accelerated the disassembly of bovine brain tubulin microtubules, increasing their shortening speeds by about

threefold, from 380±36 nm·s⁻¹ at our initial level of 1 mM magnesium to 1200±40 nm·s⁻¹ at 20 mM magnesium (*Figure 2a and b*). Consistent with our prediction, adding magnesium also increased the amplitudes of pulses measured in the wave assay (*Figure 2c*). At 2 pN of trapping force, the mean amplitude increased by 50% from 19.2±2.7 nm at 1 mM magnesium up to 29.1±2.8 nm at 20 mM magnesium (*Figure 2d*). This magnesium-dependent increase in pulse amplitude might be explained simply by lengthening the protofilament curls, as suggested by early cryo-electron microscopy studies. However, it might also reflect increases in the mechanical stiffness or curvature of the protofilaments, or in the number of protofilaments that push against the bead in the wave assay (as discussed below).

## Adding magnesium increases work output from protofilament curls

To determine whether adding magnesium affects the mechanochemical work output from curling protofilaments, we measured pulse amplitudes across a variety of trapping forces and magnesium concentrations (*Figure 3* and *Figure 3—figure supplement 1*). Measuring pulse amplitude as a function of force enables estimation of the total capacity for mechanical work output in the assay, which is given by the area under the amplitude vs force curve (*Figure 3a*; *Driver et al., 2017*). Based on a line fit to the data, we estimated work output from the bovine brain microtubules in 1 mM magnesium at 107±69 pN·nm (*Figure 3b*). Adding magnesium increased the work output monotonically, raising it to 177±0.1 pN·nm at 20 mM magnesium (*Figure 3b*). This magnesium-induced increase was mainly due to enlargement of the pulses measured at low trapping force; extrapolating the line fits to zero force suggested that the unloaded pulse amplitude (i.e. the amplitude that would be measured in the absence of opposing trap force) increased 57% from 23.3±0.9 nm at 1 mM magnesium to 36.6±0.1 nm at 20 mM magnesium (*Figure 3c*). By contrast, extrapolating the linear fits to higher forces suggested relatively little change in the maximum force at which the pulses were completely suppressed (~9 pN) (*Figure 3a*). Altogether, these observations show that magnesium increases mechanical work output from curling protofilaments while also increasing the lateral height that they project from the microtubule wall.

Notably, the mechanical work output from bovine microtubules was about threefold less than we measured previously from microtubules composed entirely of recombinant yeast tubulin under similar conditions (~300 pN·nm at 1 mM magnesium) (*Driver et al., 2017*). This difference, like magnesium-dependent differences, could reflect altered contour lengths, bending stiffnesses, average curvatures, numbers of curling protofilaments pushing on the beads, or a combination thereof.

## Curl elongation alone explains the magnesium-dependent increase in work output

To develop a deeper understanding of how magnesium increases the mechanical work output from curling protofilaments, we created a simple model of protofilament bending. The model relates structural aspects of protofilament curls, such as their relaxed curvature and the average number of dimers they contain, together with an estimate of their flexural rigidity, to predict the force-deflection behavior of a group of curls projecting radially outward from a microtubule tip. In real protofilaments, elastic bending energy can be distributed throughout the α- and β-tubulin core structures, as well as at both the inter- and intra-dimer interfaces. Rather than modeling this complexity, we placed all the compliance of the model into single bending springs located at the inter-dimer interfaces (*Figure 4a*). This simplification was important for our analyses, because it allowed data-fitting to provide good constraints on the model parameter values. (A model with more parameters would fit the data just as well or better but would not allow meaningful estimation of parameter values, due to degeneracy.) And while our model cannot address in detail how strain might be distributed across the inter- and intra-dimer interfaces (nor across the α- and β-tubulin core structures), it can describe the overall force-deflection behavior of protofilament curls, and it provides a simple way to estimate stored strain per dimer. In essence, our model convolves all the potential contributions to elastic bending strain together into a single element (an inter-dimer spring) that provides an effective flexural rigidity per dimer.

Contour shapes for the individual protofilaments were solved by balancing the external force applied at their tips with the opposing bending spring torques at each inter-dimer node (*Figure 4b*, left). To model the force-deflection behavior of a group of protofilaments, single protofilaments were arranged radially, according to a 13-protofilament geometry (*Figure 4b*, right) (*Amos and Klug,*

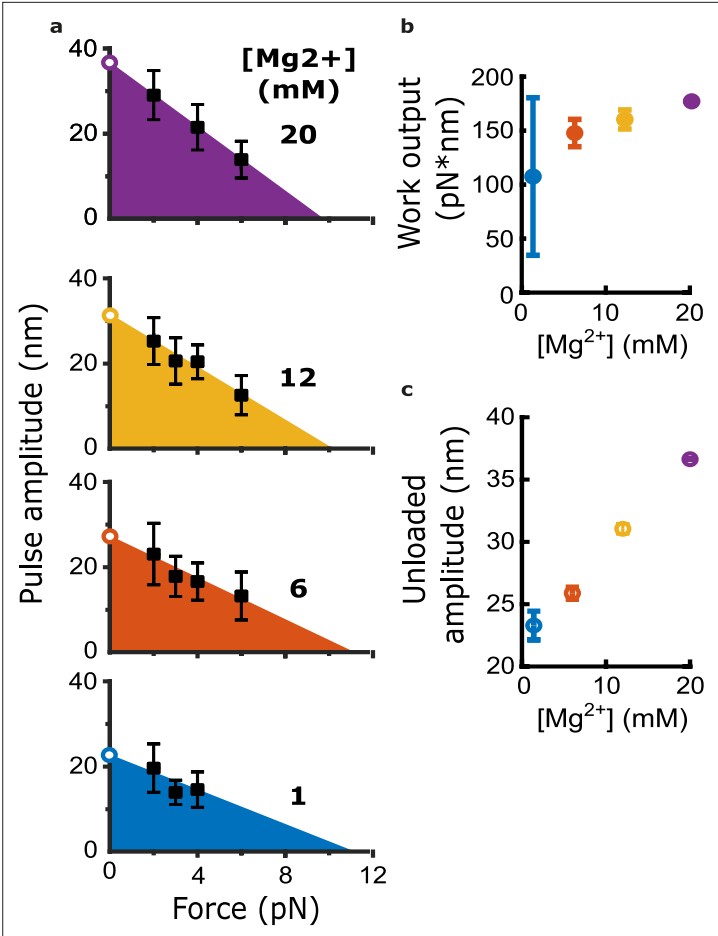

**Figure 3.** Magnesium increases the mechanical work output harnessed from curling protofilaments. (**a**) Mean pulse amplitudes (black squares) plotted against trapping force at the four indicated magnesium concentrations. Error bars represent 95% confidence intervals, defined as ± ($t$·SEM), where $t$ is drawn from Student's $t$-distribution (with $v=N–1$ degrees of freedom and N=9–43 samples per mean). The capacity of protofilament curls to perform mechanical work in the assay was estimated at each magnesium concentration by fitting the amplitude versus force data with a line and then calculating the area under the line (colored triangular areas). To estimate unloaded pulse amplitudes, the line-fits were extrapolated to the y-intercept (open circles). (**b**) Mechanical work output, based on the colored areas shown in (**a**), plotted against magnesium concentration. Error bars represent 95% confidence intervals (estimated from the best-fit parameters, as explained in Materials and methods). (**c**) Unloaded amplitudes, based on extrapolation of the line-fits in (**a**), plotted versus magnesium concentration. Error bars represent 95% confidence intervals (estimated as explained in Materials and methods).

The online version of this article includes the following source data and figure supplement(s) for figure 3:

**Source data 1.** Individual pulse amplitudes measured using bovine microtubules across different trapping forces and magnesium levels.

**Figure supplement 1.** Cumulative distributions of pulse amplitude measured with bovine tubulin at different trapping forces and magnesium levels.

**Figure supplement 2.** Statistical comparisons of estimated work outputs across different magnesium concentrations.

*1974*). The bead was modeled as a rigid, flat surface since its curvature is negligible compared to that of the microtubule tip. Prior cryo-electron tomography studies of disassembling microtubules found almost all the variation in protofilament shape to occur in the radial direction (i.e. within a plane coincident with both the relaxed contour and the long axis of the microtubule) (*McIntosh et al., 2018*). Therefore, protofilament bending in our model was limited to the radial direction. Given these assumptions, deflection of individual protofilaments varied according to their orientation relative to

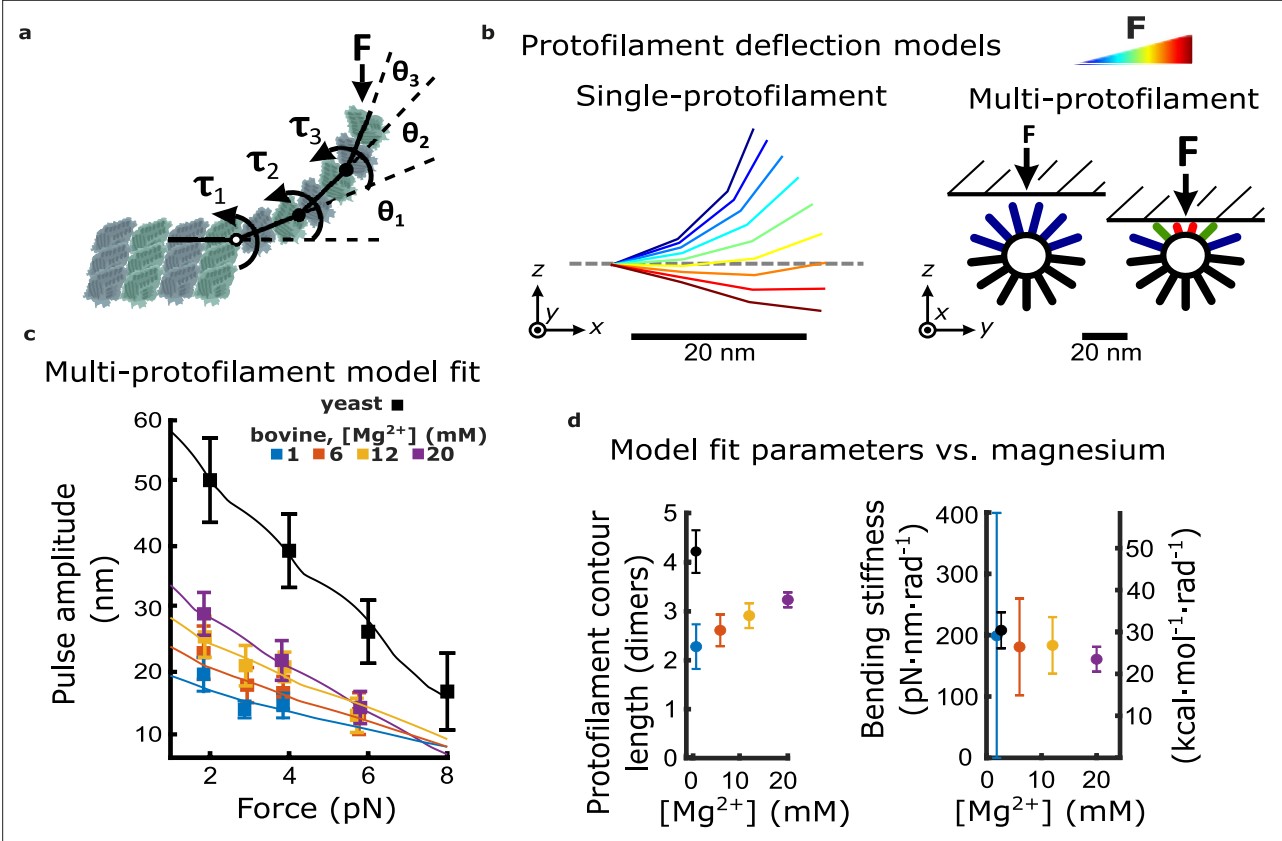

**Figure 4.** Magnesium- and species-dependent increases in work output can be explained solely by a lengthening of protofilament curls. (**a**) Model for bending of a single protofilament. Tubulin dimers are represented as rigid rods linked by Hookean torsion springs with relaxed angles of 23°. An external force, *F*, perpendicular to the microtubule long-axis, is applied at the protofilament tip. The balance between *F* and the torsion at each bending node, $\tau_n$, is used to calculate the contour shape of the protofilament (i.e. the angles $\theta_n$). (**b**) Calculated shapes for a single protofilament at different levels of external force (indicated by the color legend). Model for deflection of multiple protofilaments at a microtubule tip, seen end-on. Single protofilaments, modeled as in (**a**), are arranged radially according to the geometry of a 13-protofilament microtubule. The bead is modeled as a flat rigid surface, pushed downward onto the protofilaments to predict a force-deflection relationship. Cartoon at right shows distribution of protofilament deflections for an arbitrary bead height. (**c**) Amplitude versus force curves predicted by the multi-protofilament model, after fitting to measured pulse data (symbols) at indicated magnesium concentrations. Bovine data are recopied from *Figure 3a*. Yeast data combine new measurements with data previously published in *Driver et al., 2017*. (**d**) Two fitted parameters, the mean contour length and bending stiffness (flexural rigidity) of protofilament curls, plotted versus magnesium concentration. The fitted contour length increases with added magnesium and is larger for yeast microtubules, while the apparent flexural rigidity remains unchanged.

The online version of this article includes the following source data and figure supplement(s) for figure 4:

**Source data 1.** Individual pulse amplitudes measured using yeast and bovine microtubules across different trapping forces and magnesium levels.

**Source data 2.** Table of estimates of protofilament curvature reported in the literature.

**Figure supplement 1.** How force-deflection behavior of the single protofilament model changes with variation in the number of segments (dimers), the intrinsic bending stiffness, and the relaxed angle per tubulin dimer.

**Figure supplement 2.** Estimates of protofilament curvature from micrographs of disassembling microtubule tips presented in *Mandelkow et al., 1991*.

**Figure supplement 3.** Comparison of force-deflection relationship for a single protofilament, and multiple protofilaments arranged to reflect geometry at a microtubule tip.

**Figure supplement 4.** Comparison of disassembly speeds for bovine versus yeast microtubules.

**Figure supplement 5.** Statistical comparisons of estimated contour lengths and bending stiffnesses across different magnesium concentrations.

the bead surface (*Figure 4b*, right). A detailed analysis of changes in the force-deflection profile that occur with respect to changes in the average curvature, average dimers per curl, and flexural rigidity is shown in the supplemental material (*Figure 4—figure supplement 1*).

To fit the behavior of this multi-protofilament model to the measured pulse amplitude versus force data at each magnesium concentration, we adjusted the average number of dimers in each

curl (i.e. the curl contour length) and the stiffness of the bending springs. We kept the relaxed angle per dimer fixed at 23° because, in the absence of microtubule-associated proteins, the curvature of protofilaments at microtubule tips disassembling in vitro is consistently between 20 and 25° per dimer (*Figure 4—source data 2*), and this curvature does not change appreciably with added magnesium (*Figure 4—figure supplement 2*; *Mandelkow et al., 1991*) (nor with added calcium; *Müller-Reichert et al., 1998*). Because the bead acts as a lever, measured axial displacements of the bead are larger than the lateral deflections of the protofilaments by a leverage factor of approximately twofold (*Figure 1—figure supplement 2*; *Driver et al., 2017*). Predicted amplitude vs force curves were roughly linear, but with slight 'ripples' that occurred because movement of the bead toward the microtubule gradually engaged more protofilaments (*Figure 4c*; *Figure 4—figure supplement 3*; see Materials and methods for details). Optimal fit parameters are plotted as functions of magnesium in *Figure 4d*.

The fitted contour lengths of protofilaments increased monotonically with added magnesium, from 2.3±0.5 dimers at 1 mM magnesium to 3.2±0.2 dimers at 20 mM. However, the fitted bending stiffness per dimer, 176±15 pN·nm·rad$^{-1}$, did not appreciably change with added magnesium (*Figure 4d*). These results suggest that magnesium increases pulse amplitude and work output by lengthening the protofilament curls, without eliciting any change in their intrinsic stiffness or curvature.

## Curl elongation alone explains the larger pulses from yeast microtubules

To understand why yeast microtubules generated larger, more energetic pulses relative to bovine microtubules, we fit our multi-protofilament model to the amplitude versus force data measured from microtubules composed entirely of recombinant yeast tubulin (*Figure 4c*). As in our analysis of the bovine microtubule data, we allowed both the curl contour length and the stiffness of the bending springs to vary while keeping the relaxed angle per dimer fixed at 23°, consistent with cryo-electron tomograms of kinetochore microtubules in yeast (*McIntosh et al., 2018*). The contour length that best fit the yeast data, 4.4±0.5 dimers per curl, was 1.9-fold higher than the contour length inferred at identical magnesium concentration (1 mM) from the bovine data, 2.3±0.5 dimers per curl (*Figure 4d*). The bending stiffness per dimer that best fit the yeast data, 206±44 pN·nm·rad$^{-1}$, was statistically indistinguishable from that inferred from the bovine data (*Figure 4d*). These observations suggest that protofilament curls at yeast microtubule tips are longer but have the same intrinsic mechanical rigidity as the curls at bovine microtubule tips.

## Removing the β-tubulin tail suppresses magnesium's enhancement of disassembly speed but not of pulse amplitude

Prior studies have suggested that longer protofilament curls might arise simply as a consequence of faster disassembly speeds (*Tran et al., 1997*). Consistent with this view, when we increased magnesium from 1 to 20 mM, we observed a threefold increase in disassembly speed (*Figure 2b*) concomitant with a 1.6-fold increase in pulse amplitude (*Figures 2d and 4c*). Likewise, yeast microtubules disassembled fourfold faster than bovine microtubules at 1 mM magnesium (*Figure 4—figure supplement 4*) and generated threefold larger pulses (*Figure 4c*). Faster disassembly speeds imply that GDP-tubulins lose their lateral bonds more quickly, which equivalently can be viewed as an accelerated rate of growth of the protofilament curls at disassembling tips. However, curl size is dictated not only by curl growth but also by curl breakage; the mean steady state curl length will depend on a kinetic balance between the rates of curling and breakage (*Tran et al., 1997*) . In principle, both these rates could vary in a magnesium-dependent manner. To distinguish the potential influence of magnesium on curl breakage from its obvious effect on disassembly speed (and therefore on curl growth rate), we sought a method to slow bovine microtubule disassembly at elevated levels of magnesium. A recent discovery pointed to one such method. Fees and Moore found that removing the β-tubulin C-terminal tail, by treating microtubules with the protease subtilisin, suppresses the effect of magnesium on disassembly speed (*Fees and Moore, 2018*). Thus, at high magnesium concentration, subtilisin-treated microtubules disassemble much more slowly than untreated microtubules. If magnesium lengthens protofilament curls solely because it accelerates disassembly, then subtilisin treatment should suppress the magnesium-dependent enlargement of pulses in the wave assay.

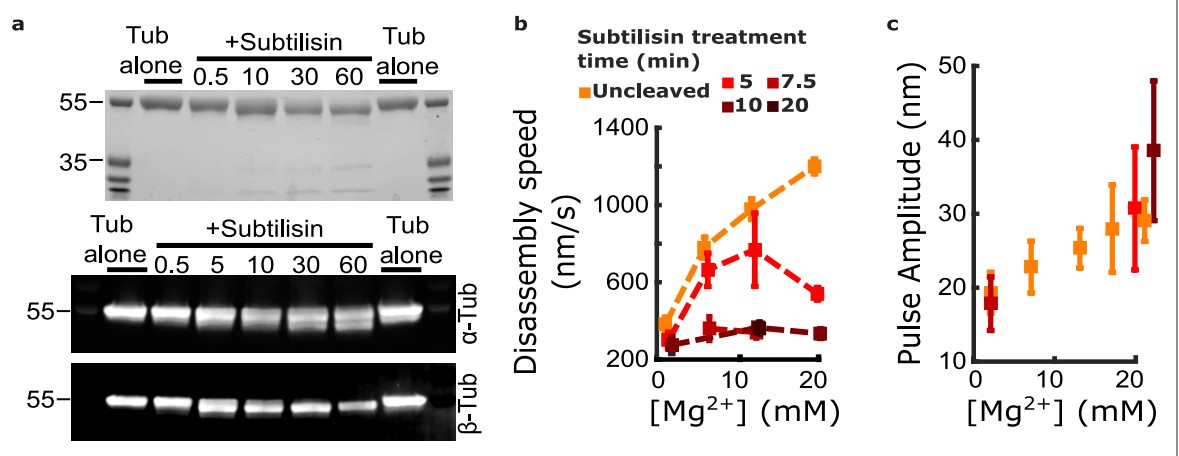

**Figure 5.** Removing the β-tubulin C-terminal tail suppresses magnesium's acceleration of disassembly speed but not its enhancement of pulse amplitude. (**a**) Proteolytic products of tubulin treated with subtilisin for the indicated times (in minutes), before quenching with 1 mM phenylmethylsulfonyl fluoride, visualized by Coomassie staining (top image) or Western blotting (bottom image). αTub DM1A:FITC mouse conjugate 1:1000 and βTub CST 9F3 rabbit 1:1000 were used for primary antibody staining. (**b**) Mean disassembly speeds, measured after treatment of tubulin with subtilisin for the indicated durations and plotted versus magnesium concentration. The large magnesium-dependent acceleration of disassembly seen with untreated tubulin (orange symbols) was suppressed after 10–20 min subtilisin treatment (dark red symbols). Error bars represent 95% confidence intervals, defined as ± ($t$·SEM), where $t$ is drawn from Student's $t$-distribution (with $v=N-1$ degrees of freedom and N=5–51 samples per mean). Data for untreated tubulin are recopied from *Figure 2b*. (**c**) Pulse amplitudes, measured in the wave assay at 2 pN trapping force after treatment of tubulin with subtilisin, plotted versus magnesium concentration. Symbol colors indicate subtilisin treatment times according to the legend of (**a**). Treatment with subtilisin did not suppress the effect of magnesium on pulse amplitude. Error bars represent 95% confidence intervals, defined as ± ($t$·SEM), where $t$ is drawn from Student's $t$-distribution (with $v=N-1$ degrees of freedom and N=28–44 samples per mean).

The online version of this article includes the following source data for figure 5:

**Source data 1.** Uncropped image of Coomassie-stained gel showing subtilisin-treated bovine tubulin.

**Source data 2.** Uncropped image of anti-alpha-tubulin Western blot of subtilisin-treated bovine tubulin.

**Source data 3.** Uncropped image of anti-beta-tubulin Western blot of subtilisin-treated bovine tubulin.

**Source data 4.** Individual pulse amplitudes and disassembly speeds measured using microtubules assembled with subtilisin-treated bovine tubulin.

**Source data 5.** Full raw unedited image of Coomassie-stained gel showing subtilisin-treated bovine tubulin.

**Source data 6.** Full raw unedited image of anti-alpha-tubulin Western blot of subtilisin-treated bovine tubulin.

**Source data 7.** Full raw unedited image of anti-alpha-tubulin Western blot of subtilisin-treated bovine tubulin.

Contrary to this prediction, however, subtilisin treatment did not reduce pulse amplitudes in the wave assay. Consistent with the prior work of Fees and Moore, we found that subtilisin treatment for 10–20 min was sufficient to remove the β-tubulin C-terminal tail (*Figure 5a*) and to suppress almost completely the magnesium-dependent acceleration of disassembly (*Figure 5b*). While disassembly of untreated control microtubules was strongly accelerated, from 380±36 to 1200±40 nm·s⁻¹, when magnesium was increased from 1 to 20 mM, the disassembly after 10 or more minutes of subtilisin treatment was consistently slower and remained at approximately 300 nm·s⁻¹ across the same range of magnesium levels (e.g. 333±24 nm·s⁻¹ at 20 mM magnesium). Despite this strikingly slower disassembly, the mean pulse amplitude measured in the wave assay after subtilisin treatment remained at least as large as that measured in controls with untreated tubulin (*Figure 5c*). At 20 mM magnesium, the mean pulse amplitude generated after 5 to 10 min of subtilisin treatment was 30.7±8.3 to 38.5±9.4 nm (respectively), a size very similar to (or even slightly larger than) the mean amplitude generated by untreated microtubules, which was 29.1±2.8 nm. This observation indicates that magnesium enlarges protofilament curls independently of its acceleration of disassembly and suggests that distinct magnesium-binding sites probably underlie these two effects.

## Discussion

### Yeast and mammalian microtubules store similar lattice strain energies

Fitting our wave assay data with the multi-protofilament model has allowed us to directly estimate a key biophysical property of curling protofilaments at disassembling microtubule tips: their flexural rigidity. Previously, this property was only inferred indirectly, from static cryo-electron tomograms (*McIntosh et al., 2018*) or from stiffness measurements of intact microtubules (*Hawkins et al., 2010*; *Kononova et al., 2014*; *Molodtsov et al., 2005*; *VanBuren et al., 2005*). Our fitted estimate for bending stiffness, 176±15 pN·nm·rad$^{-1}$, implies that fully straightening a protofilament from its relaxed curvature into a lattice-compatible state would require approximately 17 $k_B$T of work energy per tubulin dimer, or 10 kcal/mol. This represents a very substantial fraction (~80%) of the free energy available from GTP hydrolysis, ~12.3 kcal/mol (*Desai and Mitchison, 1997*; *Howard, 1996*), consistent with previous suggestions that most of the energy derived from hydrolysis is stored as curvature strain in the microtubule lattice (*Caplow et al., 1994*), and consistent with our previous lower-bound estimate (*Driver et al., 2017*). Moreover, our analysis suggests that the flexural rigidity of curling protofilaments is conserved between yeast and bovine tubulin, and therefore that the amount of strain energy stored per tubulin dimer in the microtubule lattice is probably also conserved.

The idea that protofilament flexural rigidity and stored lattice strain are conserved, despite a billion years of evolution separating yeast and vertebrates, suggests that these biophysical properties are crucial to microtubule function. Indeed, most current models assume that microtubule dynamic instability arises from the counteracting influences of lateral bonding versus lattice strain, which tend to stabilize and destabilize the polymer, respectively (*Gudimchuk et al., 2020*; *McIntosh et al., 2018*; *VanBuren et al., 2005*; *VanBuren et al., 2002*). Given the importance of dynamic instability for cell viability, there may be strong selective pressure to maintain a specific lattice strain energy.

### Protofilament curl length can affect mechanical work output

In contrast to their consistent flexural rigidity, the average length of protofilament curls at disassembling microtubule tips can vary widely depending on tubulin species and buffer conditions (*McIntosh et al., 2018*; *McIntosh et al., 2013*). By our estimates, the average curl length grew ~50% as magnesium was increased from 1 to 20 mM. And curls at yeast microtubule tips were ~twofold larger than those at bovine microtubule tips. These curl enlargements were associated with greater mechanical work output in the wave assay, as expected, since longer curls store more elastic energy and can push the bead laterally farther away from the microtubule surface. We suggest that longer protofilament curls might similarly enhance microtubule-driven motility in vivo. In budding yeast, where each kinetochore attaches a single microtubule tip (*Winey et al., 1995*), and where robust tip-coupling depends on the ring-forming Dam1 complex (*Miranda et al., 2005*; *Umbreit et al., 2014*; *Westermann et al., 2005*; *Westermann et al., 2006*), a minimum curl length might be required for microtubule-encircling Dam1 rings to efficiently harness curl energy via the conformational wave mechanism (*Molodtsov et al., 2005*). In other species whose kinetochores attach numerous microtubule tips and lack any ring-forming complexes, coupling might depend less on the conformational wave mechanism and instead might rely on biased diffusion (*Asbury et al., 2011*). An attractive idea is that the larger and more energetic pulses produced by yeast microtubules in the wave assay, as compared to bovine microtubules, might reflect stronger selective pressure to maintain long protofilament curls, because yeast might depend more heavily on long curls for mitosis.

Another possibility suggested by our work is that cells might actively tune protofilament curl properties in order to enhance microtubule-driven motility. Because free magnesium is generally thought to be buffered around 1 mM inside eukaryotic cells (*Grubbs, 2002*; *Hille, 2001*; *Romani and Scarpa, 1992*), we have viewed magnesium primarily as a biochemical tool, rather than a physiological mechanism for tuning curl properties. Interestingly, however, transient increases in free magnesium have recently been seen during metaphase and anaphase in dividing HeLa cells, where they apparently contribute to chromosome condensation (*Maeshima et al., 2018*). But the estimated concentrations remain too low (0.3–1 mM) to significantly enlarge pulses in our wave assay. Therefore, we currently favor the idea that plus end-binders (+TIPs) and other microtubule-associated proteins known to alter microtubule tip morphology (*Cassimeris et al., 2001*; *Desai et al., 1999*; *Farmer et al., 2021*; *Girão et al., 2020*; *Kerssemakers et al., 2006*) could enlarge or stiffen protofilament curls, potentially

enhancing their mechanical work output in a spatiotemporally regulated manner. We hope to investigate the effects of +TIPs on wave assay pulses in the future.

## Magnesium directly inhibits the breakage of protofilament curls

Our measurements also reveal new information about the relationship between disassembly speed and protofilament curl length, and about the mechanisms by which magnesium affects these tip properties. Classic work suggested that elongation of protofilament curls by magnesium might be a simple indirect consequence of its acceleration of disassembly (*Tran et al., 1997*). However, in 20 mM magnesium, subtilisin-treated microtubules disassembled threefold more slowly than untreated controls, and yet their pulse amplitudes remained consistently elevated, at well over 30 nm on average. Likewise, we previously showed that microtubules composed of a hyperstable T238V mutant tubulin disassemble sevenfold more slowly, and yet they generated pulses with amplitudes indistinguishable from wild-type (*Driver et al., 2017*). These observations indicate that disassembly speed and curl length are not strictly coupled, and that magnesium-dependent enlargement of protofilament curls is not simply a consequence of accelerated disassembly. Rather, magnesium must directly inhibit the breakage of protofilament curls.

The effects of magnesium on disassembly speed and on curl length appear to be mediated by different interaction sites on tubulin. Magnesium's acceleration of disassembly depends on the β-tubulin C-terminal tail, since this effect is suppressed upon removal of the β-tail by subtilisin (*Fees and Moore, 2018*). But subtilisin treatment did not suppress the effect of magnesium on pulse amplitudes in the wave assay, indicating that magnesium inhibits curl breakage through another interaction site (or sites), outside the β-tail. The C-terminal tail on α-tubulin is more resistant to subtilisin proteolysis and was left partially intact by our treatment (*Figure 5a*). Therefore, one possibility is that magnesium stabilizes protofilament curls by interacting with the α-tubulin tail. Alternatively, the effect might depend on an interaction with GDP in the exchangeable nucleotide-binding site, which is located at the inter-dimer interface. The affinity of magnesium for GDP in the exchangeable site is reportedly in the millimolar range (*Correia et al., 1987*; *Mejillano and Himes, 1991*), which is much weaker than its affinity for GTP, and near the range where we measured increased pulse amplitudes.

## Tuning curl properties could facilitate rigorous testing of their importance for kinetochore motility

The ability to tune protofilament curl properties by adjusting magnesium levels or tubulin isoforms suggests new approaches for testing the importance of curling protofilaments in kinetochore motility. If curling protofilaments exert force to drive kinetochore movement, as proposed in conformational wave-based models, then elongating the curls could enable protofilaments to push more productively against the kinetochore, potentially changing the processivity, attachment strength, or switching behavior of the kinetochore-microtubule interface. In addition, we anticipate using our wave assay and the analytical tools described here to explore other methods for modifying biophysical properties of protofilament curls. In particular, the ability to tune bending stiffness or intrinsic curvature would provide additional ways to test the importance of protofilament curls in microtubule-based motility.

# Materials and methods

**Key resources table**

| Reagent type (species) or resource | Designation | Source or reference | Identifiers | Additional information |
|---|---|---|---|---|
| Antibody | Biotinylated anti-penta-His mouse monoclonal IgG$_1$ | R&D Systems Inc. clone # AD1.1.10 | BAM050 - His Tag Biotinylated Antibody | Used for preparing anti-His beads as described under **Bead and slide preparation for wave assay**, below. |
| Other | Streptavidin coated polystyrene particles, 0.44 µm in diameter | Spherotech Inc. | SVP-05–010 | Used for preparing anti-His beads as described under **Bead and slide preparation for wave assay**, below. |

*Continued on next page*

*Continued*

| Reagent type (species) or resource | Designation | Source or reference | Identifiers | Additional information |
|---|---|---|---|---|
| Other | Biotinylated porcine brain tubulin | Cytoskeleton Inc. | T333 - Tubulin Protein (Biotin): Porcine Brain | Used for preparing coverslip-anchored microtubule seeds as described under **Bead and slide preparation for wave assay**, below. |
| Peptide, recombinant protein | Avidin | Vector Laboratories | A-3100–1 | Used for preparing coverslip-anchored microtubule seeds as described under **Bead and slide preparation for wave assay**, below. |
| Other | Biotinylated bovine serum albumin | Vector Laboratories | B-2007–10 | Used for preparing coverslip-anchored microtubule seeds as described under **Bead and slide preparation for wave assay**, below. |
| Chemical compound, drug | Odyssey Blocking Buffer | LI-COR Biosciences | 927–40000 | Used as described under **Western blotting**, below. |
| Other | Subtilisin protease | Sigma-Aldrich | P5380 - Proteinase from *Bacillus licheniformis*, Subtilisin A | Used to cleave C-terminal tubulin tails as described under **Digestion of tubulin with subtilisin**, below. |
| Antibody | Anti-β-tubulin rabbit monoclonal | Cell Signaling Technology | Anti-β-tubulin (9F3) Rabbit mAb #2128 | Used at 1:1000 as described under **Western blotting**, below. (1:1000) |
| Antibody | Anti-α-tubulin mouse monoclonal | Sigma-Aldrich clone DM1A | F2168 - Anti-α-tubulin-FITC Mouse mAb | Used at 1:1000 as described under **Western blotting**, below. (1:1000) |
| Antibody | Goat anti-rabbit polyclonal | LI-COR Biosciences | 926–32211 IRDye 800CW secondary | Used at 1:5000 as described under **Western blotting**, below. (1:5000) |
| Antibody | Goat anti-mouse polyclonal | LI-COR Biosciences | 926–68070 IRDye 800CW secondary | Used at 1:5000 as described under **Western blotting**, below. (1:5000) |

## Purification of tubulin from bovine brain

Tubulin was purified from bovine brain using two cycles of polymerization and depolymerization to a final concentration of 200 µM (*Castoldi and Popov, 2003*). Samples were frozen in liquid $N_2$ and stored at –80°C.

## Purification of recombinant His₆-tubulin from yeast

Plasmids to express wild-type yeast αβ-tubulin with a His₆ tag fused to the C-terminus of β-tubulin were previously described (*Ayaz et al., 2014*; *Ayaz et al., 2012*; *Johnson et al., 2011*). The integrity of all expression constructs was confirmed by DNA sequencing. Wild-type yeast αβ-tubulin was purified from inducibly overexpressing strains of *Saccharomyces cerevisiae* using nickel affinity and ion exchange chromatography (*Ayaz et al., 2014*; *Ayaz et al., 2012*; *Johnson et al., 2011*). Tubulin samples for the laser trap assays were prepared at UT Southwestern, aliquoted, and snap-frozen in storage buffer (10 mM PIPES pH 6.9, 1 mM $MgCl_2$, 1 mM EGTA), containing 50 µM GTP, shipped on dry ice to the University of Washington, and stored at –80°C.

## Bead and slide preparation for wave assay

To prepare anti-His beads, ~30 pM of streptavidin-coated polystyrene microspheres (Spherotech Inc, SVP-05–10) was incubated with 30 pM biotinylated anti-penta-His antibodies (R&D Systems Inc, BAM050) for 30 min, washed extensively, and then stored at 4°C for up to several months. For each experiment, a small channel ~1 mm wide was formed by bonding a KOH-cleaned glass coverslip to a clean glass slide using two parallel strips of double-stick tape. Biotinylated bovine serum albumin (Vector Laboratories, B-2007–10) was incubated in the channel for 15 min, then washed out with 80 µL warm BRB80 (80 mM PIPES, 120 mM K⁺, 1 mM $MgCl_2$, and 1 mM EGTA, pH 6.8). Avidin DN (Vector Laboratories, A-3100–1) was incubated in the channel for 5 min, then washed out with 40 µL warm BRB80. GMPCPP-stabilized, biotinylated microtubule seeds were assembled from bovine brain tubulin (*Castoldi and Popov, 2003*) and biotinylated porcine brain tubulin (Cytoskeleton, Cat #T333), and incubated in the channel for 5 min before washing with growth buffer (1 mM GTP in BRB80). Just prior to each experiment, a small aliquot of anti-His beads was pre-incubated with a mixture of plain

and biotinylated BSA (at 10 and 0.1 mg mL$^{-1}$, respectively) for 30 min, then decorated with His$_6$-yeast tubulin, and then added to growth buffer containing 10–25 µM bovine tubulin. This reaction mixture was added to the seed-decorated coverslip, the slide was then sealed with nail polish, and mounted on the optical trap. Pre-incubation of the anti-His beads with biotinylated BSA was important for preventing non-specific attachment of the beads to the microtubules. Control experiments with beads lacking anti-His antibody confirmed that the attachments were specific after the BSA pre-incubation.

To ensure that most beads attached via single antibodies, the molar ratio of antibodies to beads was kept very low, ~1:1, such that the fraction of tubulin-decorated beads that would attach to the growing end of a microtubule under manual manipulation was typically less than 10%. Active beads attached readily to growing ends but not to the sides of microtubules. Their exclusive preference for growing ends is expected because the anti-His antibodies on the bead become quickly occupied by individual, unpolymerized tubulin dimers upon initial mixing with the His$_6$-tagged yeast tubulin. Laterally attached beads, which are required for the wave assay, arise by the incorporation of bead-tethered yeast tubulin dimers into the growing ends of microtubules (which can be composed either of bovine or yeast tubulin, depending on the experiment) followed by polymerization of the microtubules past the beads. A detailed protocol for slide preparation is given in our recent publication (*Murray et al., 2022*).

## Trapping instrument

The optical trap instrument used for this assay has been described in previous work (*Franck et al., 2010*). The instrument was based around a Nikon inverted microscope (TE2000) with a Nikon 100×1.4 NA oil Plan Apo IR CFI objective. A 1064 nm Nd:YVO4 laser (Spectra Physics J20-BL10-106Q) was used as a trapping beam, focused at the center of the field of view. A 473 nm laser (LaserPath Technologies, DPSS-473–100) was used as a microtubule cutting beam, focused into an ellipse at an intermediate distance between the trap center and edge of the field of view. Both lasers were actuated by shutters (Vincent Associates, VS25S2ZMO). Microtubules and beads were visualized by video enhanced differential interference contrast (VE-DIC), with illumination by a mercury arc lamp (X-Cite 120) and accomplished through two standard Wollaston prisms and polarizers (*Walker et al., 1988*). Motion control and force-feedback were implemented through servo-control of a three-axis piezo stage with internal capacitive position sensors (Physik Instrumente, P-517.3CL) and a piezo controller (Physik Instrumente, E-710). Custom software written in LabVIEW (National Instruments) was used for instrument control and data acquisition. The source code is publicly available at https://github.com/casbury69/laser-trap-control-and-data-acquisition (*Asbury, 2021*). Briefly, analog signals from the position sensor were sampled at 40 kHz using an analog-to-digital conversion board (National Instruments, PCI-6251). Commands were sent to the piezo stage controller through a GPIB digital interface (National Instruments, GPIB-USB-B). Both the bead and stage positions were downsampled to 200 Hz for file storage.

## Measurement of pulses driven by protofilament curling

Suitable beads laterally attached to coverslip-anchored microtubules were identified. Suitable microtubules were firmly anchored by one end to the slide surface, and able to freely rotate about their surface anchor, without other interfering microtubules bundled alongside or crossing along their length. To establish the initial loaded state, the laterally attached bead was trapped, and the microtubule and bead were pulled in the opposite direction of the tether, toward the cutting laser location. The beads were raised slightly above the coverslip surface to ensure the surface did not interfere with measurement. The force clamp was initiated, and microtubule depolymerization was triggered by trimming off the stabilizing cap using the cutting laser. Position signals from the trapped bead were recorded using the force clamp software (described above under Trapping instrument), including the static baseline position and the pulse driven by protofilament curling motion. Candidate pulses were evaluated for inclusion in data analysis on the basis of their amplitude relative to the standard deviation of the baseline noise; a detection threshold of three times the standard deviation was used to accept or reject pulses. For most records this threshold was 6–10 nm, as detailed in our prior publication (*Murray et al., 2022*). The fraction of events that yielded measurable pulses in bovine microtubule experiments was lower than in yeast microtubule experiments mainly because the pulses were smaller on average, and therefore more of them fell below our detection threshold.

Average pulse amplitudes are reported in the text as mean ± 95% confidence interval, which was estimated as ± ($t$·SEM), where $t$ is drawn from Student's $t$-distribution using $v=N-1$ degrees of freedom. The number of samples per mean ranged from N=9 to 43, as indicated in the legends of *Figures 2 and 3*. All the individual pulse amplitude values are included in excel spreadsheet format as supplemental source data files. They are also publicly available as a MATLAB (Mathworks) data file at https://github.com/protofilamentdude/Protofilament-Bending-Models (*Murray, 2022*). Uncertainties in the measured work outputs shown in *Figure 3b* were estimated by propagating uncertainty in the parameters of the best-fit lines of *Figure 3a* through the calculation of area under each line. (Uncertainties in the best-fit line parameters were assumed to be uncorrelated.)

## Measurement of microtubule disassembly speeds

Slides for measuring disassembly speeds were prepared as described above (in Bead and slide preparation for wave assay) but without the addition of yeast-tubulin decorated beads. Microtubules were visualized by VE-DIC and recorded using a digital video disc recorder (Toshiba, DR430). The stabilizing GTP-caps of microtubules were trimmed off using laser scissors to induce disassembly. Disassembly speeds of individual microtubules were measured using imageJ and mTrackJ (*Meijering et al., 2012*).

## Multi-protofilament model

Single protofilaments were modeled as a series of rigid rods linked by Hookean bending springs with an angular spring constant, $\kappa$, a non-zero relaxed angle, $\theta_i$ (*Figure 4a*), and a segment length, $r$=8.2 nm. A downward force at the protofilament tip was balanced by the torque at each spring node to yield a system of nonlinear equations (see *Equations 1–3* below for a 4-node, 3-segment system).

$$\tau_3 = Fr\cos\left(\theta_1 + \theta_2 + \theta_3\right) - \kappa\left(\theta_3 - \theta_i\right) \tag{1}$$

$$\tau_2 = Fr\left[\cos\left(\theta_1 + \theta_2\right) + \cos\left(\theta_1 + \theta_2 + \theta_3\right)\right] - \kappa\left(\theta_2 - \theta_i\right) \tag{2}$$

$$\tau_1 = Fr\left[\cos\left(\theta_1\right) + \cos\left(\theta_1 + \theta_2\right) + \cos\left(\theta_1 + \theta_2 + \theta_3\right)\right] - \kappa\left(\theta_1 - \theta_i\right) \tag{3}$$

The system of non-linear equations was solved numerically for the angle at each node ($\theta_1$, $\theta_2$, $\theta_3$,...) using a variant of the Powell dogleg method (*Powell, 1970*), for a range of forces and a given set of parameters $\kappa$ and $\theta_i$. Using the angles ($\theta_1$, $\theta_2$, $\theta_3$,...), the deflection of the protofilament tip was calculated at each force. This method was repeated for modeled protofilaments of lengths 1–5 segments.

The force-deflection relationship for multiple protofilaments at a microtubule tip interacting with a bead was calculated as follows: the bead was assumed to be an infinitely flat, rigid surface because the 440 nm beads used in experiments were nearly 20-fold larger in diameter than the microtubules. Protofilaments were assumed to be distributed radially about the microtubule axis in a 13-protofilament configuration. Bending was only allowed in the plane traversed by the microtubule axis and the protofilament axis because it has been observed in electron microscopy that such a plane includes most deviations in protofilament position (*McIntosh et al., 2018*). The position of the bead surface was varied from where it contacted the most apical (upward pointing) protofilaments, down to the microtubule wall. Accordingly, groups of protofilaments were engaged sequentially based on their distribution around the microtubule tip (*Figure 4b*, right). This sequential engagement of protofilaments manifested as slight ripples in the force-deflection curve and changed slightly depending on the rotational angle of the microtubule tip (*Figure 4—figure supplement 3*). To consider a variety of possible rotational angles, the force-deflection curves for the two microtubule tip rotations depicted in *Figure 4—figure supplement 3* were averaged together prior to fitting. The model was implemented with custom code written in MATLAB (Mathworks) that is publicly available at https://github.com/protofilamentdude/Protofilament-Bending-Models (*Murray, 2022*).

## Multi-protofilament model fitting

To fit the multi-protofilament model to the pulse amplitude versus force data, the amplitude data was first converted to bead-to-microtubule surface height, assuming a 36 nm tether length (*Figure 1—figure supplement 2*). The model was fit to the data using a Levenberg-Marquardt nonlinear least-squares algorithm with inverse-variance weights, yielding the fitted force-deflection relationship, and parameters for the stiffness per tubulin dimer and the average contour length. 95% confidence intervals were calculated using the Jacobian for each parameter. Fitting was performed using custom code

written in MATLAB (Mathworks) and available publicly at https://github.com/protofilamentdude/Protofilament-Bending-Models (*Murray, 2022*).

## Digestion of tubulin with subtilisin

To cleave the C-terminal tails from tubulin, bovine brain tubulin was thawed quickly, and mixed to a final concentration of 100 μM with 1% subtilisin (Sigma Aldrich P5380) in a buffer containing 1 mM GTP, 8 mg/mL BSA, 80 mM PIPES, pH 6.8, 1 mM $MgCl_2$, 1 mM EGTA, and immediately placed at 30°C. To halt the cleavage reaction, phenylmethylsulfonyl fluoride was added to a final concentration of 1 mM, and the cleavage product placed on ice.

## Western blotting

Samples were run on a 7.5% Bis-Tris SDS-PAGE gel and transferred to polyvinylidene difluoride membranes. The membranes were blocked for 1 hr with 1:1 1× PBS and Odyssey Blocking Buffer (Licor Biosciences, 927–40000;). Primary and secondary antibodies were diluted in 1:1 1× PBS and Odyssey Blocking Buffer. The blots were incubated for 1 hr with the following primary antibodies: βTub (Cell Signaling Technology, 9F3) rabbit 1:1000, αTub DM1A:FITC (Sigma, F2168) mouse conjugate 1:1000. The blots were washed 3× with PBST (1× PBS and 0.01% Tween-20). The blots were incubated with secondary antibodies diluted in 1:1 1× PBS and Odyssey Blocking Buffer, IRDye 680RD goat anti-mouse IgG secondary antibody (Licor, 926–68070) 1:5000, IRDye 800CW goat anti-rabbit IgG secondary antibody (Licor, 926–32211) 1:5000. The blots were washed 3× with PBST and 1× with PBS, then imaged with a Licor Odyssey DLx imaging system. Images were adjusted for brightness and contrast with Image-J software.

## Data availability

All data generated and analyzed during this study are included in the manuscript and supporting files. Source data files are provided with all the individual wave amplitude values and disassembly speeds for *Figures 1–5* and their supplements. A source data file with all the individual wave amplitude values is also publicly available at https://github.com/protofilamentdude/Protofilament-Bending-Models (*Murray, 2022*), which includes custom MATLAB (Mathworks) code for fitting these data with the multi-protofilament model. The custom LabView (National Instruments) code that we use for control of our laser trapping instruments is publicly available at https://github.com/casbury69/laser-trap-control-and-data-acquisition (*Asbury, 2021*).

## Acknowledgements

We are grateful for critical reading of the manuscript and feedback from Trisha N Davis, Linda Wordeman, Joshua D Larson, Bonnibelle K Leeds, and John J Correia. This work was supported by NIH grants to CLA (R35GM134842) and LMR (R01GM098543), by a Packard Fellowship to CLA (2006–30521), and by a grant from the Robert A Welch Foundation to LMR (I-1908). The authors declare no competing financial interest.

---

## Additional information

### Funding

| Funder | Grant reference number | Author |
| --- | --- | --- |
| National Institutes of Health | R35GM134842 | Charles L Asbury |
| National Institutes of Health | R01GM098543 | Luke M Rice |
| Packard Foundation | 2006-30521 | Charles L Asbury |
| Welch Foundation | I-1908 | Luke M Rice |

| Funder | Grant reference number | Author |
|---|---|---|

The funders had no role in study design, data collection and interpretation, or the decision to submit the work for publication.

## Author contributions

Lucas E Murray, Conceptualization, Data curation, Software, Formal analysis, Validation, Investigation, Visualization, Methodology, Writing - original draft, Writing – review and editing; Haein Kim, Conceptualization, Investigation, Methodology; Luke M Rice, Conceptualization, Resources, Funding acquisition, Methodology, Writing – review and editing; Charles L Asbury, Conceptualization, Software, Supervision, Funding acquisition, Methodology, Project administration, Writing – review and editing

## Author ORCIDs

Lucas E Murray ⓘ http://orcid.org/0000-0001-5680-732X
Luke M Rice ⓘ http://orcid.org/0000-0001-6551-3307
Charles L Asbury ⓘ http://orcid.org/0000-0002-0143-5394

## Decision letter and Author response

Decision letter https://doi.org/10.7554/eLife.83225.sa1
Author response https://doi.org/10.7554/eLife.83225.sa2

# Additional files

## Supplementary files

• MDAR checklist

## Data availability

All data generated and analyzed during this study are included in the manuscript and supporting files. Source data files are provided with all the individual wave amplitude values and disassembly speeds for Figures 2–5 and their supplements.

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
