## [Editor Report]

This important and technically sophisticated work advances our understanding of force production by depolymerizing microtubules with implications for the generation of forces that segregate chromosomes during cell division. The authors present compelling evidence for their mechanistic conclusions. This work will be of interest for cell biologists and biophysicists interested in cell division and force production by biopolymers.

---

## [Decision Letter]

**Decision letter after peer review:**

Thank you for submitting your article "Working strokes produced by curling protofilaments at disassembling microtubule tips can be biochemically tuned and vary with species" for consideration by *eLife*. Your article has been reviewed by 3 peer reviewers, one of whom is a member of our Board of Reviewing Editors, and the evaluation has been overseen by Suzanne Pfeffer as the Senior Editor. The following individuals involved in review of your submission have agreed to reveal their identity: Jeffrey K Moore (Reviewer #2); Arne Gennerich (Reviewer #3).

Essential revisions:

1) The reviewers noted technical differences between the previous experiments using yeast microtubules (Driver et al. 2017) and the experiments reported here using mammalian microtubules. Please see major point of reviewer 1. It will be important to discuss whether/how these differences may affect the comparison of the properties of microtubules from the two different species.

2) Please provide estimates for errors of the parameter values stated in the main text. See comment of referee 3.

*Reviewer #2 (Recommendations for the authors):*

1) One major conclusion of the study is that bovine microtubules generate less work output than yeast microtubules. How comparable are the experiments combining bovine and his-tagged yeast tubulin to the previous experiments with pure yeast tubulin? My understanding is that the previous experiments used yeast tubulin that was all his-tagged, providing a high density of potential attachment sites for the bead on the microtubule lattice. In contrast, the new experiments use bovine microtubules and beads sparsely decorated with his-tagged yeast tubulin and rely on the incorporation of that tubulin into the bovine microtubule lattice. It seems like the density of potential attachment sites must be much lower in the new experiments. Does this alone lead to differences in the compliance or deflection of the bead in the experiment? The authors note that in the bovine microtubule experiments, only 59% of disassembly events lead to measurable pulses at 2pN trapping force. This appears to be much lower than the previous work with yeast tubulin, which reported measurable pulses for 90% of disassembly events at <5pN. The density of attachment sites and how that might impact the deflection measurements should be considered.

2) The demonstration that magnesium can modulate the amplitude of bead deflection and work output by depolymerizing ends is an important contribution of the study. However, while this may be a useful tool in vitro, the authors do not address how this might be related to biology in cells. While exploring this experimentally is beyond the scope of this study, it would be helpful if the authors included some discussion of how the magnesium concentrations used here relate to magnesium concentrations in cells. Do cells experience changes in intracellular magnesium during the cell cycle, or in different cell types or organisms, that would approach the range used here?

3) Are the work output estimates in Figure 3b different as a function of magnesium? It is not clear how the error bars are generated or if any statistical analysis was performed. The figure legend refers to the methods, but I could not find the details there.

4) The description of the model depicted in Figure 4 could be clearer. Specifically, the tubulin dimers are said to be modeled as a rigid rod, but later in the paragraph the text describes measuring strain at the interdimer and intradimer interfaces. Is the intradimer interface then modeled as flexible or as part of a rigid rod? It seems more appropriate to model it as flexible, since tubulin heterodimers are curved in solution and likely re-visit that curvature when they are in protofilaments that are unconstrained by lateral binding to other protofilaments.

5) That removing the β-tubulin tail separates magnesium's effect on depolymerization rate from its effect on bead deflection is a very interesting and surprising result. I would have expected those to be linked. I'm left wondering how magnesium exerts these two effects and whether they might be connected through the β tubulin tail? Relevant to this study, I'm curious whether magnesium could act by linking curled protofilament neighbors together and generate greater bead deflection through additive forces from multiple protofilaments pushing on the bead? Figure 5c shows that pulse amplitude appears to be higher but also increasingly variable as magnesium increases. This indicates that there may be different classes of protofilaments pushing on the bead at high magnesium. Does longer protofilament length sufficiently explain this variability?

*Reviewer #3 (Recommendations for the authors):*

As the presented work is carried out with great care and rigor, I have no further comments besides that the authors should provide errors for the parameters in the main text. I, therefore, recommend the publication of this study.

---

## [Author Response]

Essential revisions:1) The reviewers noted technical differences between the previous experiments using yeast microtubules (Driver et al. 2017) and the experiments reported here using mammalian microtubules. Please see major point of reviewer 1. It will be important to discuss whether/how these differences may affect the comparison of the properties of microtubules from the two different species.

Please see our detailed response in Reviewer #1’s Public Review.

2) Please provide estimates for errors of the parameter values stated in the main text. See comment of referee 3.

Please see our detailed responses to Reviewers #2 and #3.

Reviewer #2 (Recommendations for the authors):1) One major conclusion of the study is that bovine microtubules generate less work output than yeast microtubules. How comparable are the experiments combining bovine and his-tagged yeast tubulin to the previous experiments with pure yeast tubulin? My understanding is that the previous experiments used yeast tubulin that was all his-tagged, providing a high density of potential attachment sites for the bead on the microtubule lattice. In contrast, the new experiments use bovine microtubules and beads sparsely decorated with his-tagged yeast tubulin and rely on the incorporation of that tubulin into the bovine microtubule lattice. It seems like the density of potential attachment sites must be much lower in the new experiments. Does this alone lead to differences in the compliance or deflection of the bead in the experiment? The authors note that in the bovine microtubule experiments, only 59% of disassembly events lead to measurable pulses at 2pN trapping force. This appears to be much lower than the previous work with yeast tubulin, which reported measurable pulses for 90% of disassembly events at <5pN. The density of attachment sites and how that might impact the deflection measurements should be considered.

As mentioned above, we apologize for not making it clearer in our original manuscript that the beads in both scenarios, with bovine and yeast microtubules, were tethered to microtubules by a single antibody to the same C-terminal tail of yeast β-tubulin. Thus, the differences in pulse amplitude cannot be explained by differences in the tethering. In our revised manuscript, we now mention explicitly in Results that the beads were tethered by single antibodies (lines 95 to 100). In Methods we significantly expanded the section about preparation of beads and how they became tethered (lines 365 to 393).

The fraction of events that yielded measurable pulses in bovine microtubule experiments was lower than in yeast microtubule experiments mainly because the pulses were smaller on average, and therefore more of them fell below our detection threshold, which was typically around 6 to 10 nm. This is now explained in Methods (lines 424 to 428).

2) The demonstration that magnesium can modulate the amplitude of bead deflection and work output by depolymerizing ends is an important contribution of the study. However, while this may be a useful tool in vitro, the authors do not address how this might be related to biology in cells. While exploring this experimentally is beyond the scope of this study, it would be helpful if the authors included some discussion of how the magnesium concentrations used here relate to magnesium concentrations in cells. Do cells experience changes in intracellular magnesium during the cell cycle, or in different cell types or organisms, that would approach the range used here?

This is an interesting question, and we thank you for raising it. Free magnesium is generally thought to be buffered around 1 mM inside cells, so we have been viewing magnesium primarily as a biochemical tool for altering protofilament properties rather than a physiological mechanism. Classic studies found large, cell cycle-dependent increases in total magnesium in yeast, just prior to cell division (e.g., Walker 1980 *J Cell Sci*). Unfortunately, they could not distinguish free magnesium from magnesium bound tightly to nucleic acids or other cellular constituents, which probably represents most of the total. One interesting modern study, using fluorescence- and FRET-based probes, reported a transient increase in free magnesium during metaphase and anaphase in HeLa cells (Maeshima 2018 *Curr Biol*). But the concentrations remained too low (0.3 to 1 mM) to significantly enlarge pulses in our wave assay. Therefore, we currently favor the idea that protofilament curls in cells might be enlarged or stiffened by +TIPs, or by other microtubule-associated proteins (MAPs) that alter tip morphology. In the Discussion section of our revised manuscript, we now mention the likely buffering of free magnesium around 1 mM inside cells. We cite the Maeshima 2018 paper as an interesting counterexample of cell cycle-dependent regulation of free magnesium, albeit at concentrations too low to affect pulse amplitudes in our wave assay (lines 304 to 311). The paragraph ends with our favored idea that +TIPs or MAPs could tune protofilaments in cells to enhance their mechanical work output.

3) Are the work output estimates in Figure 3b different as a function of magnesium? It is not clear how the error bars are generated or if any statistical analysis was performed. The figure legend refers to the methods, but I could not find the details there.

Thank you for pointing out this omission. In the revised version, our method for estimating uncertainties in the work output estimates of Figure 3b is described in Methods (lines 434 to 437). We also added a supplemental table of p-values indicating the statistical significance for all possible pairwise comparisons between the four work output estimates of Figure 3b (Figure 3 —figure supplement 2).

4) The description of the model depicted in Figure 4 could be clearer. Specifically, the tubulin dimers are said to be modeled as a rigid rod, but later in the paragraph the text describes measuring strain at the interdimer and intradimer interfaces. Is the intradimer interface then modeled as flexible or as part of a rigid rod? It seems more appropriate to model it as flexible, since tubulin heterodimers are curved in solution and likely re-visit that curvature when they are in protofilaments that are unconstrained by lateral binding to other protofilaments.

Thank you for this suggestion to improve the clarity of our model description. In retrospect, we agree that our original description was confusing, especially regarding how the intra-dimer interface was handled in the model. In our revision, we have extensively rewritten the corresponding section of Results (lines 169 to 181). We now clarify that all the compliance of the model was placed into bending springs located at the inter-dimer interfaces. We explain how this represents a simplification relative to the real situation, where elastic strain energy is undoubtedly distributed throughout the α- and β-tubulin core structures and at both the inter- and intra-dimer interfaces. We further explain why we chose to convolve all these potential contributions to elastic strain into a single element (a single inter-dimer bending spring). Doing so makes the model simple enough for data-fitting to provide meaningful constraints on parameter values, and yet the model can describe overall force-deflection behavior for protofilament curls, and it provides an estimate of stored strain per dimer.

5) That removing the β-tubulin tail separates magnesium's effect on depolymerization rate from its effect on bead deflection is a very interesting and surprising result. I would have expected those to be linked. I'm left wondering how magnesium exerts these two effects and whether they might be connected through the β tubulin tail? Relevant to this study, I'm curious whether magnesium could act by linking curled protofilament neighbors together and generate greater bead deflection through additive forces from multiple protofilaments pushing on the bead? Figure 5c shows that pulse amplitude appears to be higher but also increasingly variable as magnesium increases. This indicates that there may be different classes of protofilaments pushing on the bead at high magnesium. Does longer protofilament length sufficiently explain this variability?

We fully agree that it is very surprising that magnesium’s effect on disassembly speed is separable from its effect on wave amplitude. The β-tail is clearly important for magnesium-dependent acceleration of disassembly, as your lab first showed, and we have now repeated. Given that wave amplitudes remain magnesium-sensitive and high even after subtilisin-removal of the β-tail, we suggest that another magnesium-binding site must be responsible for its effect on pulse amplitude. However, at this point we can only speculate about where that site might be located. In principle it might be possible to identify the key site(s) by generating point mutants and testing these in the wave assay. This would require a large amount of work that in our opinion would go far beyond the scope of this paper.

Regarding possible lateral association between curling protofilaments, we think changes in such lateral associations are less likely than changes in protofilament contour length to explain the magnesium- and species-dependent differences we measured in pulse amplitude, for several reasons (also mentioned above in our response to Reviewer #1): First, there is good evidence for lengthening of protofilament curls at disassembling tips (e.g., Mandelkow 1991, Tran and Salmon 1997), but we are not aware of convincing evidence for magnesium or species-dependent increases in the propensity of curling protofilaments to remain laterally associated. Second, an increase in lateral association should increase the effective flexural rigidity of the curls, but under all the conditions we examined, pulse enlargement was associated with a steepening of the amplitude-vs-force relation – i.e., with softening, not stiffening. Our model indicates that this softening can be fully explained by an increase in protofilament contour length, without any change in the intrinsic flexural rigidity of the protofilament curls.

We do not completely understand all the sources of variability in our measured pulse amplitudes, but we note that the magnitude of this variability seems to scale (at least roughly) with mean pulse amplitude. This trend is qualitatively what we would expect from a simple model with stochastic breakage of independent protofilaments (like the model of Tran and Salmon 1997). However, we would need much more data to accurately measure the variability and test rigorously whether such a simple model can explain it. This could be an interesting avenue for future study.

Reviewer #3 (Recommendations for the authors):As the presented work is carried out with great care and rigor, I have no further comments besides that the authors should provide errors for the parameters in the main text. I, therefore, recommend the publication of this study.

Thank you for the suggestion to provide estimates of uncertainty for the values we report in the main text. In our revised version, we now report all values given in the text as means ± 95% confidence intervals.